# INVESTIGATING GROKKING PHENOMENA BELOW THE CRITICAL DATA REGIME

## ABSTRACT

In this paper, we investigate the phenomenon of *grokking*, wherein models exhibit delayed generalization following overfitting on training data. Our focus is on studying grokking in data regimes where the amount of training data is below the critical threshold necessary for grokking to occur naturally. We examine several scenarios that provide insight on the grokking phenomenon and suggest avenues for practical applications. We first consider training with a strong regularizer, specifically Knowledge Distillation(KD) from a model that has grokked on a distribution ($p_1$) to induce grokking on a different distribution ($p_2$). We find that this can lead to much faster grokking and reduced critical data size. Furthermore, we show that reducing the weight norm, a key focus in previous grokking studies, is not a necessary condition for grokking. We next, explore the scenario where we aim to train a larger size model on a joint distribution ($p_1$, $p_2$). We demonstrate that achieving generalization under the critical data size is not possible through standard supervised training. However, we show that we can achieve generalisation if we first perform grokking on two models with the individual distributions and distill this result into the larger model. Finally we consider a continual pretraining setup, where a grokked model transitions from distribution $p_1$ to $p_2$, we find that KD from the grokked model leads to faster generalization, even when the available data constitutes as little as $10\%$ of the dataset. This is noteworthy because generalization might otherwise be unattainable in such low-data conditions. Moreover, distillation mitigates catastrophic forgetting of previously learned knowledge. Our analysis offers new insights on the grokking phenomenon when knowledge transfer is feasible and illustrates the substantial role KD can play in accelerating generalization especially under low-data regime.

## 1 INTRODUCTION

In the rapidly evolving landscape of machine learning, the ability of models to adapt and generalize across varying data distributions (Singh et al., 2024b;a; Van de Ven & Tolias, 2019; Fang et al., 2020; Liang et al., 2024) remains a paramount challenge. Traditional training paradigms often struggle in dynamic environments where data distributions shift or where data is scarce, leading to models that either fail to generalize or require extensive computational resources to retrain. Recently, the phenomenon of grokking (Power et al., 2022) has demonstrated new perspectives on the generalization behaviour and how a model can transition to a perfect generalization after long episode of overfitting and pure memorization (Arpit et al., 2017). Many recent studies attempted at providing a better understanding of grokking, attributing it to weight decay that steers the optimization towards generalization zone even after reaching a zero loss on the training data (Ishida et al., 2020). Grokking is predominantly observed in low-data regimes; however, it has been shown that beyond a critical data threshold, grokking cannot occur.

To the best of our knowledge, grokking has only been studied in the context of a single training distribution, primarily focusing on weight decay as its underlying cause. In this work, we explore grokking in the data regimes lower than critical data, and systematically analyze the influence of grokked models on related varying distributions in conditions that trigger grokking.

Specifically we address the following questions

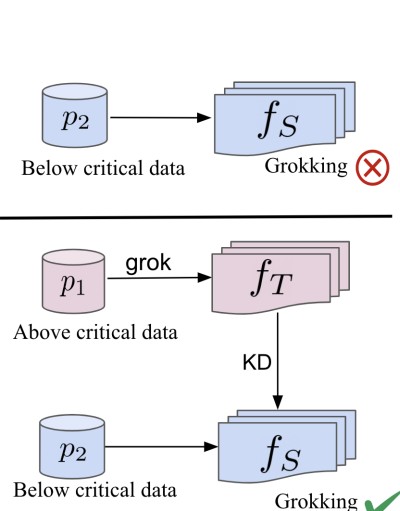

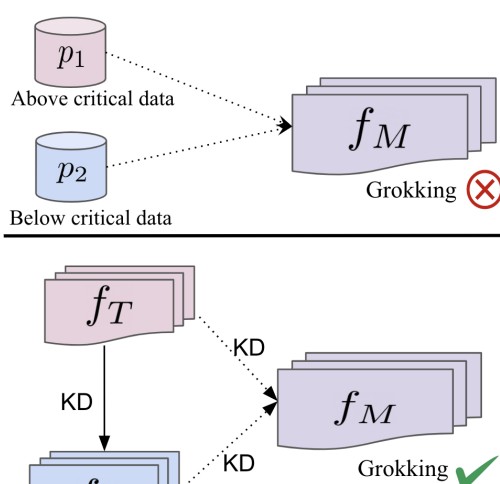

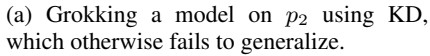

(a) Grokking a model on $p_2$ using KD, which otherwise fails to generalize.

(b) Distilling from multiple grokked models $f_T$, $f_S$ yields grokking on a larger model $f_M$ below critical data.

Figure 1: Fig 1a demonstrates that $f_S$ successfully groks below the critical data size when trained using KD from an already grokked model $f_T$ which otherwise fails to generalize on its own. In Figure 1b, a larger model $f_M$, tasked with jointly learning $p_1$ and $p_2$, fails to generalize when either dataset falls below the critical size. However, distilling knowledge from the smaller grokked models $f_S$ and $f_T$ enables $f_M$ to grok, allowing it to generalize effectively even when data is below the critical threshold.

**Q-1**: Can we leverage an already grokked model in *learning* another model especially on a varying distribution?

**Q-2**: Is it possible to observe grokking when the available data is *less* than the *critical amount*?

**Q-3**: Are weight decay and decreasing weight norms the *sole drivers* of the grokking phenomenon?

To address the first question, we conducted extensive experiments by initially training a 1 layer Transformer (Vaswani et al., 2017) model to grok on a distribution $p_1$. This model serves as the Teacher ($f_T$), which is then used to train a Student model ($f_S$) on a different distribution $p_2$. We observed that not only does the Student model $f_S$ exhibit grokking on the new distribution $p_2$, but with distillation, the number of steps required to achieve grokking are also reduced. This approach is especially relevant in data crunch situations where the availability of data for $p_2$ is limited. By utilizing a pre-grokked model on $p_1$ we aim to facilitate rapid adaptation to $p_2$, thereby mitigating the challenges posed by data scarcity. A natural question arises, *why differ distributions?* The primary reason is to give a flavour of practical utility, where a perfectly generalizable model can be used to assist other models in transferring knowledge under distribution shift. Our investigation is motivated by the pressing need for models that can seamlessly transition between different data distributions without incurring prohibitive computational costs. This is can be highly useful in various domains like continual learning, multi-task learning, domain generalization, etc.

To address the second question, our investigation reveals that Knowledge Distillation(KD) offers multiple advantages, one of which is reducing the number of iterations required for grokking. Utilizing KD, we empirically demonstrate that grokking can occur even when the amount of data is less than the critical data size. The critical data size as defined in (Liu et al., 2022b; Varma et al., 2023) is the minimum amount of data below which generalization is impossible.

Previous studies have established that grokking occurs within specific data regimes. For instance, Power et al. (2022) mentions that for large dataset sizes, training and validation losses track each other closely. Similarly, Nanda et al. (2023) observes that with sufficient data, the gap between training and test loss vanishes. Varma et al. (2023) further investigate the behavior of learning curves around the critical dataset size, identifying various manifestations of grokking. In contrast, our experiments demonstrate that grokking can be observed even below the critical data regime, highlighting the efficacy of KD in facilitating generalization under limited data conditions.

Finally, in addressing the third question, we empirically demonstrate that generalizing solutions do not always lie on smaller weight norm spheres in parameter space, contrary to the arguments presented in (Liu et al., 2022b; Varma et al., 2023).

Nanda et al. (2023) propose that training can be divided into three phases: memorization of the training data, circuit formation (where the network learns a mechanism that generalizes), and cleanup (where weight decay removes the memorization components). They further suggest that the sudden transition to perfect test accuracy in grokking occurs during the cleanup phase, after the generalizing mechanism has been learned. Through our rigorous experiments, we refute these ideas. We consistently demonstrate examples of grokking occurring with zero weight decay and an increase in parameter weight norm across different training settings, thereby ruling out these factors as the primary reasons or explanations for grokking.

To substantiate our claims, we conduct a series of experiments across various algorithmic tasks, including addition and subtraction. These tasks provide a controlled environment to rigorously evaluate the efficacy of our proposed methodologies. The results from these experiments underscore the potential of grokking-based approaches in enabling efficient model training under constraints of dynamic data distributions and limited data availability. Our findings contribute a comprehensive framework for developing more robust and adaptable machine learning systems, paving the way for advancements in fields where data variability and scarcity are prevalent.

## 2 RELATED WORK

**Grokking** was first observed for algorithmic datasets by (Power et al., 2022). Since then considerable efforts have been made to understand grokking.

**Theoretical explanation of Grokking on simpler networks:** Rubin et al. (2024) provides analytical predictions from a first-order phase transition perspective on feature learning and demonstrate a mapping between Grokking and the theory of phase transitions. Similarly Levi et al. (2024) provided explicit analytical solutions for the training loss, generalization loss and accuracy dynamics in a linear network. Analysing polynomial regression using a two-layer neural network Kumar et al. (2024) hypothesized that grokking may arise from a transition from lazy to rich learning regime. Lyu et al. (2024) suggest that the sharp transition in test accuracy may stem from a dichotomy of implicit biases between the early and late training phases.

**Empirical demonstrations of Grokking:** Humayun et al. (2024) explains that grokking materializes in a wide range of practical settings, such as training of a convolutional neural network (CNN) on CIFAR10 (Krizhevsky, 2012). They introduce the new concept of delayed robustness, whereby a deep neural network groks adversarial examples and becomes robust, long after interpolation and/or generalization. Notsawo Jr et al. (2023) proposed to predict grokking using the spectral signature from the Fourier transform to detect specific oscillations in the early training phase. Liu et al. (2022a) attributes grokking to the slow formation of good representations owing to the presence of four learning phases: comprehension, grokking, memorization, and confusion. They find representation learning to occur only in a "Goldilocks zone"(including comprehension and grokking) between memorization and confusion. Nanda et al. (2023) demonstrated that grokking, rather than being a sudden shift, arises from the gradual amplification of structured mechanisms encoded in the weights, followed by the later removal of memorizing components. This process is followed by the systematic elimination of memorization components. Barak et al. (2022) suggests that generalization is due not to random search, but to hidden progress of SGD to gradually amplify a Fourier gap. Thilak et al. (2022) links grokking to the "Slingshot mechanism" marked by cyclic transitions between stable and unstable training

**Relationship of Grokking and Dataset Size:** Varma et al. (2023) employed circuit efficiency analysis to reveal that generalization is slower to learn but more efficient. They also introduced a concept of 'critical data size' below which it is extremely easy to memorise the training dataset, without generalisation. Training with these data points will result in suboptimal test loss (i.e., semi-grokking). And fine-tuning grokked models with smaller data sizes will lead to poor test performance (i.e., ungrokking). Doshi et al. (2023) indicated that regularization methods could correct errors in the training samples. Liu et al. (2022b) analyzed the loss landscapes of neural networks in explaining many aspects of grokking: data size dependence, weight decay dependence, emergence of representations

**Knowledge Distillation(KD):** Knowledge distillation Hinton (2015) is a widely used technique for model compression Sun et al. (2019); Sarfraz et al. (2021); Mishra & Marr (2017), building more efficient neural network families (Huang et al., 2017; Singh et al., 2024a;b), quantizing existing networks to use fewer bits for weights and activations Wu et al. (2016) and distilling knowledge from larger networks into smaller ones (Tung & Mori, 2019). The method involves training a smaller student model to replicate the behavior of a larger teacher model. This approach has been applied successfully in various domains, including natural language processing and computer vision. Our work builds on this foundation by focusing on task-level knowledge transfer in algorithmic tasks with varying data distributions.

# 3 EXPERIMENTAL SETUP

We trained a decoder only transformer to perform experiments on algorithmic tasks of the form $((a@b)\%P)$, where @ represents operator for any of the binary operations. In this work, we focus on addition and subtraction tasks. Our choice of algorithmic data is based on previous studies (Nanda et al., 2023; Varma et al., 2023; Liu et al., 2022b; Power et al., 2022; Liu et al., 2022a) have consistently demonstrated the phenomenon of grokking on these tasks. By utilizing these well-established benchmarks, we are able to derive significant insights into the underlying mechanisms of grokking, which can inform our understanding of more complex and practical applications.

The input to the model is of the form $[a, b, @, P]$, where we read the output of the task $c$ from the last token $P$. In our primary experiments, each binary arithmetic modulo $P$ task is referred as $p_1$ for a specific prime number $P$. A distribution shift is introduced by changing the $P$, while keeping the task operation same. For example, consider algorithmic addition modulo $P$ task: $((a + b)\%P)$. For a given prime $P = P1$, the distribution is referred to as $p_1$, whereas for some other $P = P_2 \neq P_1$, the distribution is referred to as $p_2$. Our results are consistent regardless of the choice of $P_1$ and $P_2$.

For an Input Space: $\mathcal{X} \subseteq \mathbb{R}^d$, Output Space: $\mathcal{Y} = \{1, 2, \ldots, P\}$ we have a general definition for Data Distribution as $\mathcal{D}$ over $\mathcal{X} \times \mathcal{Y}$. The loss function without KD is the Cross Entropy given as:

$$L_{\text{CE}}(\theta) = \mathbb{E}_{(x,y)\sim\mathcal{D}} \left[ -\log f_S^y(x; \theta) \right] \tag{1}$$

where $f_S(\cdot; \theta)$ :, is the Student Network: parameterized by $\theta$.

For knowledge distillation, we use Kullback-Leibler (KL) Divergence Loss:

$$L_{\text{KL}}(\theta) = \mathbb{E}_{x\sim\mathcal{D}_X} \left[ D_{\text{KL}} \left( q_T(x) \| q_S(x; \theta) \right) \right] \tag{2}$$

where $D_{\text{KL}}(p\|q) = \sum_{i=1}^{K} p_i \log \left( \frac{p_i}{q_i} \right)$. This takes softened outputs as $q_T(x) = \text{softmax} \left( \frac{f_T(x)}{t} \right)$, and $q_S(x; \theta) = \text{softmax} \left( \frac{f_S(x;\theta)}{t} \right)$ where $f_T$ :, represents the Teacher model, and $t > 0$ is the Temperature used to soften probabilities.

The total distillation loss is therefore realised as:

$$L(\theta) = (1 - \alpha)L_{\text{CE}}(\theta) + \alpha L_{\text{KL}}(\theta) \tag{3}$$

where $\alpha$ controls the proportion of each loss component.

We start training by utilising only $30\%$ of the training set, to first observe grokking. We then consistently lower the data fraction to $20\%$ and $10\%$, which are below critical data regime for algorithmic addition and subtraction task as given by (Varma et al., 2023). For demonstrating the efficacy of our distillation method and to negate the dependency of weight norm and weight decay theories, we

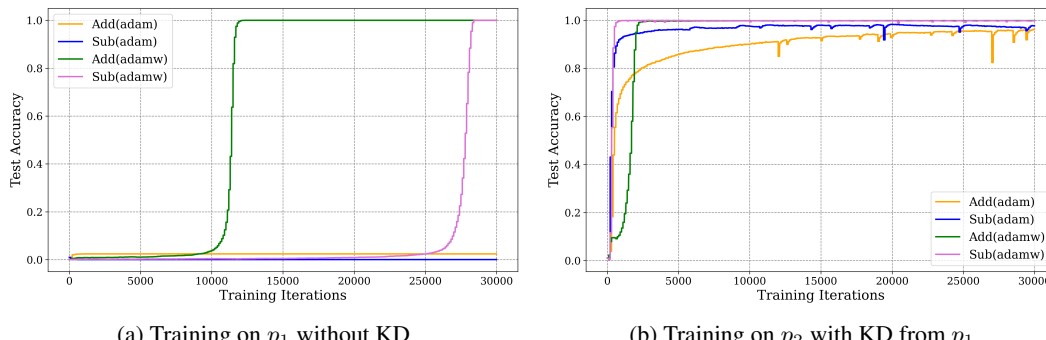

(a) Training on $p_1$ without KD

(b) Training on $p_2$ with KD from $p_1$

Figure 2: Fig 2a shows the typical grokking phenomena on distribution $p_1$ on 30% of training data, without KD. We observe that weight decay is helpful in showing grokking but its not the only underlying cause. When trained with Adam, grokking is not observed within 30000 iterations. This concurs with (Power et al., 2022). However Fig 2b demonstrates a $Student$ model trained on a different distribution $p_2$ with same fraction, but now with KD from the $Teacher$ model trained on $p_1$(Fig 2a). Distillation takes place on probability outputs from the operator token, and not the $P$ token, since we aim to learn generic operator level representations, rather than overall task level representations, which would depend on the choice of $P$. This shows that irrespective of the choice of optimizer, KD is sufficient to grokk a model, which is not dependent on weight norm or weight decay.

compare both Adam without weight decay & AdamW(with weight decay) optimizer (Loshchilov, 2017) with a learning rate $\gamma = 1e - 3$. For AdamW we set the weight decay parameter $\lambda = 1$. We perform 30,000 epochs of training with a batch size of 2048 on NVIDIA V100 GPU.

## 4 IS GROKKING TRULY DEPENDENT ON THE PARAMETER WEIGHT NORM OR WEIGHT DECAY?

We first train a 1 layer Transformer model($f_T$) on 30% of training data $p_1$. As seen in Figure 2a, grokking is observed within 30000 iterations. We further observe that weight decay helps in reducing the number of iterations as shown in (Power et al., 2022). But we observe that weight decay is not the only cause of grokking. Teacher model $f_T$, which has grokked on distribution $p_1$, can be leveraged to train a student model $f_S$ from scratch on a different distribution $p_2$, using the same fraction (30%) of $p_2$. As shown in Fig 2b, distillation from $f_T$ not only enables $f_S$ to grok on $p_2$, but also significantly accelerates the grokking process, regardless of the optimizer used. This demonstrates a practical utility of grokked models, illustrating their effectiveness in training models on varying distributions through KD. It is important to note that distillation occurs on the probability outputs from the operator token rather than the $P$ token. This approach aims to learn generic operator-level representations instead of task-specific representations, which would depend on the choice of $P$.

We observe that utilizing KD significantly reduces the number of steps required to achieve grokking, irrespective of the optimizer employed. It has been shown to provide multiple benefits in improving training dynamics. Menon et al. (2021) provided a statistical perspective on distillation, that providing the true class-probabilities from the teacher model can lower the variance of the student objective, and thus improve performance. Further Phuong & Lampert (2019) provides a generalization bound that establishes fast convergence of the expected risk of a distillation-trained linear classifier. It can be inferred from these studies (Tang et al., 2020; Cho & Hariharan, 2019; Yuan et al., 2020) that KD brings the following advantages towards training dynamics,

- **Regularization Effect through Label Smoothing:** KD smooths the labels, which acts as a regularizer and prevents overfitting.

- **Domain Knowledge Injection:** The teacher model imparts class relationships that shape the geometry of the student's logit layer.

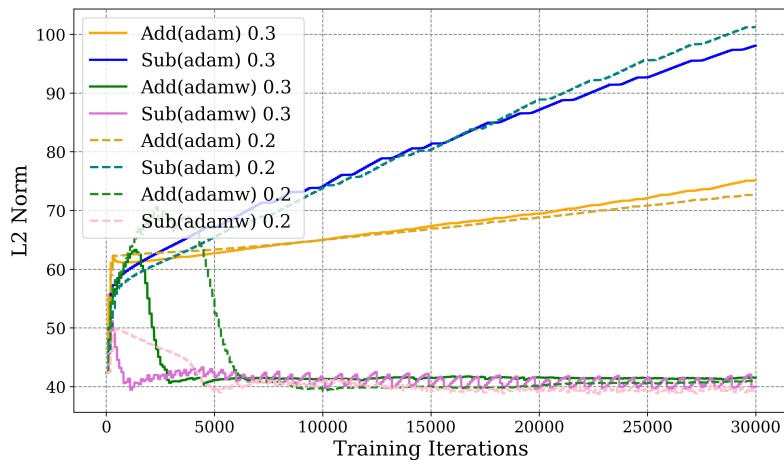

Figure 3: Evolution of the L2 weight norm for $Student$ model $f_S$ trained with Adam(without weight decay) and AdamW(with weight decay) on different fractions of $p_2$ distribution. $f_S$ is trained via KD from a grokked model $f_T$. Notably, training without weight decay the $L_2$-weight norm increases continuously, while giving generalised solutions. This rules out the necessity of decreased weight norm condition for exhibiting grokking given by (Liu et al., 2022b; Varma et al., 2023; Nanda et al., 2023)

- **Instance-Specific Knowledge:** The teacher adjusts the student model's per-instance gradients based on the difficulty of each sample, facilitating more effective learning.

Additionally, as illustrated in Figure 3, the weight norm continuously increases for both addition and subtraction tasks, yet grokking still occurs. These findings challenge the theories proposed by (Nanda et al., 2023) who suggest that the abrupt transition to perfect test accuracy during grokking occurs in the cleanup phase (where weight decay removes memorization components), following the establishment of the generalizing mechanism. Our empirical evidence contradicts these claims by demonstrating grokking even without weight decay and with increasing weight norms.

Similarly Liu et al. (2022b) induce grokking by increasing the initial weight norm and conclude that generalizing solutions lie on smaller norm spheres in parameter space. While we acknowledge that an initially higher weight norm can facilitate grokking, our results indicate that generalizing solutions do not necessarily lie on smaller norm spheres. Our modular arithmetic tasks serve as counterexamples, where the final generalizing solutions exhibit larger parameter weight norms than their initial states, and grokking occurs without the application of weight decay.

Furthermore Varma et al. (2023) claim that the transition from memorizing to generalizing circuits occurs because the generalizing circuit is more "efficient" than the memorizing circuit, in the sense that it can produce equivalent loss with a lower parameter norm. In contrast, our studies show that modular arithmetic tasks can achieve generalizing solutions with higher parameter norms without any weight decay, disproving the necessity of norm reduction for grokking.

Therefore we assert that ***neither parameter weight decay nor decreasing weight norm during optimization is inherently fundamental to observing grokking***, as highlighted by the above previous studies on modular arithmetic tasks.

## 5 IS IT POSSIBLE TO OBSERVE GROKKING BELOW CRITICAL DATA REGIME?

Building upon the observations from the previous section, where we found that KD significantly accelerates grokking at a data fraction of $30\%$, a pertinent question arises: ***Can KD facilitate generalization below this critical data threshold?*** To investigate this, we replicate the experiments described in Section 4, this time employing a reduced data fraction of $20\%$.

As illustrated in Figure 4, our results reveal that without KD, no generalization is achieved within 30,000 iterations, regardless of weight decay. This lack of generalization persists even when weight

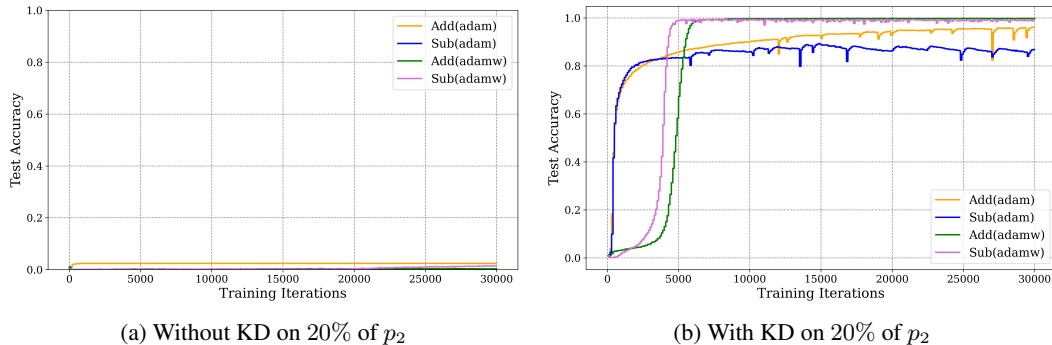

(a) Without KD on 20% of $p_2$       (b) With KD on 20% of $p_2$

Figure 4: Fig 4a demonstrates that its impossible to observe grokking when the data fraction goes below a certain critical threshold(20%.) In such a case, the model does not learn anything regardless of the optimizer. In Fig 4b, it can be clearly seen that with KD, grokking is observed for all tasks, even without weight decay. However we notice that weight decay helps in achieving a better generalisation.

decay is applied, highlighting the limitations of traditional optimization techniques in low-data regimes. In stark contrast, the application of KD enables grokking at the lower data fraction of 20%. Remarkably, even in these scenarios, the weight norm continues to increase, thereby supporting our earlier assertion that neither weight decay nor weight norm reduction is essential for the emergence of grokking.

These findings highlight the critical role of a grokked Teacher model, especially in data-scarce environments where the available training data falls below the threshold necessary for grokking or any generalisation. By leveraging a grokked Teacher model through KD, we not only accelerate the grokking process but also extend its applicability to situations with limited data. This demonstrates the practical utility of grokked models in facilitating efficient training across varying data distributions, thereby offering a robust solution for scenarios where data is constrained.

Extending the previously discussed concepts, we conducted an additional experiment by checkpointing the grokked models for different fractions $(0.3, 0.2, 0.1)$ of $p_2$ trained via distillation as discussed in previous Section 4 and Section 5. Specifically, we refer the model trained on distribution $p_1$ using 30% of the training data as $f_{p_1}$, and the grokked models trained on different fractions of distribution $p_2$ with KD as $f_{p_2}$. Our objective now becomes to train a larger transformer model capable of generalizing across both distributions $p_1$ and $p_2$. To achieve this, we compared two distinct training scenarios, as illustrated in Figure 5.

**Joint Training on Limited Data:** The larger model was trained jointly on 30% of $p_1$ and different fractions of $p_2$. In this scenario, we observed that the larger model failed to generalize when the data for $p_2$ falls below the critical size, indicating that the scarcity on any distributions impeded its ability to learn a robust and generalizable representation.

**Training via KD Only:** We conducted two sets of experiments. In the first, as shown in Figure 5, a larger model $f_M$ was trained solely through KD using the pre-trained models $f_{p_1}$ and $f_{p_2}$, without applying any cross-entropy minimization. Distillation occurred over the probability logits from the final $P$ token, as the goal was to generalize across both tasks simultaneously. In a similar setup, we performed another experiment using two grokked models, $f_{p_1}$ and $f_{p_2}$, each trained on 30% of their respective data. $f_M$ was again trained exclusively via distillation from these grokked models, but with varying fractions of both $p_1$ and $p_2$, as illustrated in Figure 6.

Remarkably, $f_M$ exhibited grokking behavior ***only when trained via KD***, even when either $p_1$ or $p_2$ was below the critical data size, successfully generalizing despite the limited data. This demonstrates that KD over the joint distribution $(p_1, p_2)$ provides a more informative signal than training with ground-truth labels. KD-enabled training allows grokking to emerge even when the data is below the critical size. Notably, this effect holds true even when the grokked teacher model $f_{p_2}$ was trained on a similarly small fraction of $p_2$ data, but with distillation from $f_{p_1}$.

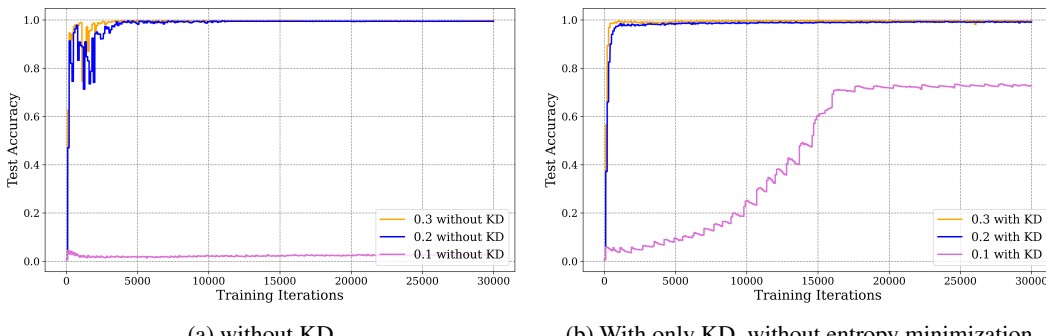

(a) without KD                          (b) With only KD, without entropy minimization

Figure 5: Performance comparison of training strategies for a larger transformer model $f_M$ on distributions of $p_1(30\%)$ and different fractions $(0.3, 0.2, 0.1)$ of $p_2$. Figure 5a shows the Joint Training regime. it can be observed that the model fails to generalise via cross entropy minimization when the training data from any of distributions falls below critical threshold. On the contrary, training a larger model alone with distillation with just $10\%$ induces grokking as shown in Figure 5b. Although we observe that when data is so scarce($10\%$), the generalization accuracy falls short of unity, because of the imperfect $f_{p_2}$, trained in data crunch situation with distillation. In a it looks like an immediate generalization for $0.2$ and $0.3$, with no grokking.

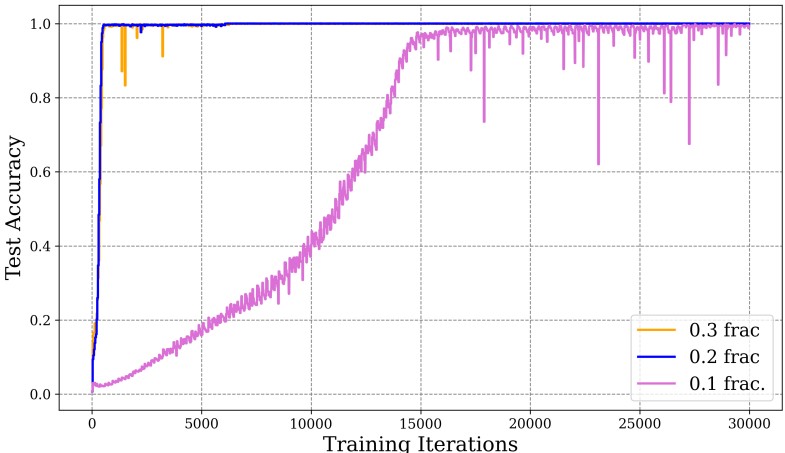

Figure 6: Training of a larger model $f_M$ via distilling from grokked models $f_{p_1}$ and $f_{p_2}$. These small models are grokked on $30\%$ of training data each. Training of larger model $f_M$ is trained with different fractions$(0.3, 0.2, 0.1)$ of $p_1$ and $p_2$, with only distillation from grokked models $f_{p_1}$ and $f_{p_2}$.

In a similar setup based on upon recent advancements in continual pretraining methodologies (Ke et al., 2023), we conducted a comprehensive experiment to evaluate the efficacy of continual pretraining transitions from a previously grokked model generalized on $p_1$ to $p_2$. Specifically, we investigated the role of KD in mitigating catastrophic forgetting during this transition. Our experimental setup involved initializing the pretraining process with a model that had achieved generalized performance on $p_1$ through grokking. We then proceeded to pretrain the model on $p_2$ under two distinct conditions: with and without the application of KD.

The results demonstrated that in the absence of KD, the model experienced almost instantaneous and severe forgetting of the previously acquired knowledge on $p_1$. Despite this rapid forgetting, the model exhibited swift generalization capabilities to the new distribution $p_2$. In stark contrast, when KD was employed during continual pretraining, the model retained nearly perfect test accuracy on $p_1$ while simultaneously achieving rapid generalization on $p_2$. Importantly, the incorporation of KD effectively prevented the occurrence of grokking, as delayed generalization was not observed in either scenario. These findings highlight the critical role of KD in preserving previously learned

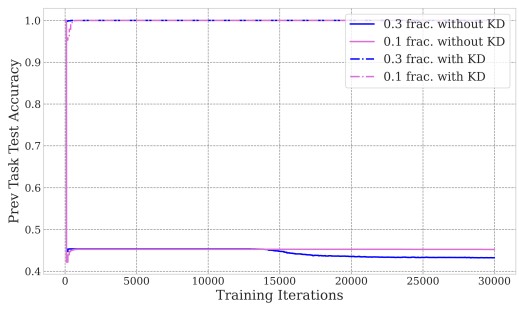 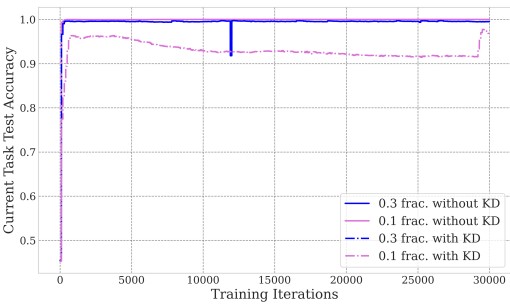

(a) Previous Task Accuracy for different fractions of data, with and without KD.

(b) Current Task Accuracy for different fractions of data, with and without KD.

Figure 7: This demonstrates continual pretraining where the grokked model on $p_1$ is continually pretrained on $p_2$. It can be clearly inferred that without KD, the performance on the previous task deteriorates rapidly, while generalising rapidly on the current $p_2$. Fig 7b shows that distillation preserves current task accuracy as well as mitigates catastrophic forgetting. Its interesting to note that training on current task from a grokked model, achieves quick generalisation without grokking. However in Fig 7b for data regime less than critical size, we observe a sudden phase transition from an already high accuracy of around $92\%$ to unity at around $28K$ steps.

information during continual pretraining. By effectively balancing the retention of legacy knowledge with the acquisition of new skills, KD serves as a robust mechanism to enhance model stability and performance in dynamic learning environments.

These results highlight that KD can facilitate generalization even in scenarios with severely limited data from multiple distributions. This is particularly pertinent in practical situations where acquiring sufficient data is challenging due to constraints such as security protocols, privacy regulations, and other restrictive factors. In such contexts, leveraging KD from pre-trained grokked models emerges as an elegant and effective solution to overcome the limitations imposed by scarce data availability.

Furthermore in all the above experiments, the consistent increase in weight norm despite successful grokking challenges existing theories that posit weight norm reduction as a fundamental driver of grokking. Our experiments provide compelling evidence that alternative mechanisms, such as the transfer of learned representations via KD, play a more pivotal role in enabling generalization under reduced data conditions. This insight opens new avenues for research into the underlying factors that contribute to grokking, moving beyond traditional optimization paradigms.

## 6 CONCLUSIONS AND FUTURE WORK

This study advances our understanding of the grokking phenomenon by exploring its behavior below critical data regime. Unlike prior research that primarily focused on a single training distribution and the influence of weight norm and weight decay, our work broadens the scope by systematically investigating how grokking can be induced with KD without relying only on weight decay and decreasing weight norms. Our findings challenge the prevailing notion that weight decay and decreasing weight norms are the sole drivers of grokking. Through rigorous experimentation, we demonstrated that grokking can occur even in the absence of weight decay and with increasing weight norms, thereby refuting earlier hypotheses that linked grokking exclusively to these factors. Additionally, we established that KD not only accelerates the grokking process but also enables generalization below the previously identified critical data threshold even in varying distributions which is significant for scenarios characterized by data scarcity, where traditional training methods falter.

Future work may extend these insights to more complex and diverse real-world tasks, further elucidate the underlying mechanisms of grokking, and explore additional strategies to harness pregrokked models for various transfer learning applications. By continuing to unravel the intricacies of grokking, we can pave the way for the development of machine learning models that not only generalize effectively but also adapt swiftly and efficiently to the ever-changing landscapes of real-world data.

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
