# OpenReview forum: "Investigating Grokking phenomena below the Critical Data Regime"
_ICLR.cc/2025/Conference — ICLR 2025 Conference Withdrawn Submission_

### Official Review · Reviewer_VSq8 · 2024-10-27

**Soundness:** 2
**Presentation:** 2
**Contribution:** 2
**Rating:** 3
**Confidence:** 4

**Summary:**

The paper provides empirical evidence challenging the notion that weight decay is the sole regularization method capable of achieving grokking in deep learning models. It demonstrates that generalization—where performance improves significantly after overfitting—can be attained through other regularization techniques. Specifically, the authors use knowledge distillation from previously grokked models as an additional loss term to regularize their model. Despite the model's weights increasing over successive iterations, grokking still occurs, countering the traditional view that weight decay alone drives this phenomenon.

**Strengths:**

1. The paper is somewhat well-written although there is significant room for improvement.

2. Some of the experiments (Figure 3) are intuitive.

**Weaknesses:**

1. The description of the experiments are not clear.

2. Loosely uses mathematical terminologies without properly defining them.

3. Experiments done only on a particular dataset.


4. No theoretical guarantee provided.

**Questions:**

I have several concerns regarding the experiments and the math notations:
1. Section 3: the authors use P for modulo operation and use small p for distribution notation which is confusing. In line 196, they describe the input space whereas they describe the distribution before that. The distribution is also confusing. What is the random variable here? How do the input data live in d dimensional space as mentioned in Line 196. They use the term `distribution' loosely everywhere.

2. Figures: it seems the model's accuracy is at 0% before grokking which is confusing to me. What is the chance accuracy here? Did the model learn anything at all? I would change the color scheme in Figure 3. It took me a lot of effort to distinguish among the shades and linestyles of green.

3. Section 4: I would elaborate more on the bullet points regarding the advantages of KD and how they relate to the current work. Referring to other works is unhelpful.

4. The paper shows empirical evidence on a selected dataset. As they do not have any theoretical guarantee, without elaborate experiments on a broad range of datasets, raises concerns for overfitting to a particular case.

5. I did not understand the experiment description in Page 7. The statement in Line 375 goes against their Figure 1 (b) top right panel. what is $f_M$ tested on in all the plots?

6. Continual learning experiment: The terminology `pretraining' is confusing for continual learning (CL) setup. They consider a narrow setup with two tasks only. The experiment description is not clear and I could not understand their hypothesis and finding after investing a significant time to grok it. CL setups are not useful unless they have a lot of tasks (~100-1000) in the learning environment. We want to look at the long term behavior of the CL algorithms. CL systems are complex and there many performance statistics to consider while proposing a CL setup. I would solely focus on the single task experiments and address the CL setup in a different paper.

7. What is the impact of similarity between p1 and p2 on the grokking? I would explore the effect of task similarity and review the transfer learning literature.

8. The author only mentions percentage of total data in experiments everywhere without mentioning the total sample size. Without knowing the actual sample size, it is impossible to know how low that data percentage is.

---

### Official Review · Reviewer_PY2d · 2024-10-28

**Soundness:** 2
**Presentation:** 2
**Contribution:** 2
**Rating:** 3
**Confidence:** 3

**Summary:**

This paper studies grokking in data regimes where the amount of training data is below the critical threshold necessary for grokking to occur naturally.  The authors conduct grokking experiments with a knowledge distillation (KD) objective, and reveal the following findings: (1) training a student model by KD from a grokked (teacher) model can accelerate grokking and reduce critical data size needed for grokking. (2) reducing weight norm is not a necessary condition for grokking. (3) KD enables generalization when available data is below the critical data size in two scenarios.

**Strengths:**

1. The topic in understanding grokking is timely and important.
2. Considering knowledge distillation setup in grokking is novel, to the best of the reviewer’s knowledge.

**Weaknesses:**

The reviewer has the following major concerns, and therefore leans on a rejection.

1. Conclusion not very surprising: regarding the finding-(1) in the above summary, when distilling a grokked model trained on a task p1 to a student model training on another task p2, it is shown that the student model is easier to generalize (acceleration of grokking) and the required data for p2 is below critical data size. However, the authors use tasks whose differences are only up to the modulus P (if the reviewer understands correctly). It is intuitive that such a KD process can inject some bias (or act as some ‘pre-training’) that facilitates the generalization on a similar task p2. Did the authors try to transfer to other tasks such as changing the binary operators?
2. Part of the conclusion is not new: regarding the finding-(2) in the above summary, existing works already showed with counterexamples that a decreasing weight norm may not be causally related to grokking, see examples in [1][2].
3. The experimental setup lacks justification:
    - the reviewer is confused about the experimental setup, specifically why the training is performed in 30000 epochs and the data fraction is 30%/20%/10%? Are these used in some prior works?
    - In section 5, the authors show that training a ‘larger’ model on a joint distribution of two tasks does not lead to grokking, but training two models on each task individually and distilling the two models into a ‘larger’ model allows for grokking. Why should one care about this setup? What is its implication?
4. The writing can be improved.
    - For example, figure 1 is not mentioned in the text if the reviewer is not mistaken. In addition, many parts in the paper miss critical citations for a smooth reading (e.g. row 48, 50-51, 201-207)
    - More intuition or explanation on why a set of experiments is conducted and why the use of KD objective makes a difference, would be greatly helpful.
5. Practical implication: the paper is motivated to consider a low-data regime (subject to security protocols and privacy regulations) for the purpose of facilitating generalization under limited data conditions, however, the reviewer had a hard time connecting the toy setting with the real-world applications. Specifically, the authors seem not to verify that a grokked model can reduce the delay generalization on the student model for a broad range of tasks.

[1] Grokking as the transition from lazy to rich training dynamics. Kumar et al. ICLR 2024

[2] Progress Measures for Grokking on Real-world Tasks. Golechha. HiLD 2024: 2nd Workshop on High-dimensional Learning Dynamics

**Questions:**

Please see the above ‘weakness’ section and clarify if there is any misunderstanding from the reviewer.


Minor:

1. At row 97, should it be ‘post-grokked’ or ‘grokked’ model instead of ‘pre-grokked’ model?
2. At row 184, it is implied that P is a constant modulus, but at row 190, P is referred to as the last token of the model’s output, which is not consistent.

---

### Official Review · Reviewer_2PyV · 2024-10-31

**Soundness:** 2
**Presentation:** 2
**Contribution:** 3
**Rating:** 5
**Confidence:** 3

**Summary:**

This paper studies the root causes and data regime that steer the occurance of grokking through empirical results. They conclude several interesting propositions and highlight the importance of knowledge distillation for grokking.

**Strengths:**

- This paper provides several insightful and counterintuitive propositions for grokking, which are helpful for this field.

**Weaknesses:**

- The empirical study only focuses on modulo operation. This is limited to derive a general conclusion. They should weaken their claims in the title and main text. Alternatively, more types of tasks should be evaluated to support these claims.
- The distribution discrepancy is not explicitly specified. Intuitively, very large distribution discrepancy cannot lead to grokking.  For  modulo operation, we can change the value of $P$. However, for other tasks, a general depiction should be included.
- The ground-truth of critical data size is not discussed. To make the experiment reliable, this value is very important. If not, even lowering the datasize, it may be still above the ground-truth critical data size.
- Necessary theoretical insights are missing. They are helpful for us to better understanding these interesting propositions.
- There exist some typos in writting.

**Questions:**

In Figure 2a, why is the test acc consistently near to zero? Although only using the 30% of training data, no any accuracy increasement is shocked for me. Could you please explain this result?

---

### Official Review · Reviewer_k8Ne · 2024-11-10

**Soundness:** 2
**Presentation:** 2
**Contribution:** 1
**Rating:** 3
**Confidence:** 4

**Summary:**

This paper provides an intriguing exploration into grokking, specifically focusing on scenarios where the data size is below the critical threshold typically necessary for this phenomenon. The authors tackle several questions on generalization through grokking under low-data conditions and demonstrate that knowledge distillation (KD) can significantly accelerate grokking.

**Strengths:**

- The study questions the necessity of weight decay and low weight norms for grokking, suggesting new insights into generalization.
- Paper is well-written

**Weaknesses:**

- The relevance of the research question is unclear. The authors fail to motivate the problem (research Q1-Q3) explored in the paper.
- It is also unclear why grokking is studied for the problem of knowledge distillation.
- Testing on real-world datasets would strengthen the practical applicability of the analysis.
- Experiments with more complex models could better validate the findings' scalability.

**Questions:**

NA

---

### Note · Authors · 2024-12-02

I have read and agree with the venue's withdrawal policy on behalf of myself and my co-authors.